# Toll-like Receptors and Cytokine Modulation by Goat Milk Extracellular Vesicles in a Model of Intestinal Inflammation

**DOI:** 10.3390/ijms241311096

**Published:** 2023-07-04

**Authors:** Chiara Grazia De Ciucis, Floriana Fruscione, Livia De Paolis, Samanta Mecocci, Susanna Zinellu, Lisa Guardone, Giulia Franzoni, Katia Cappelli, Elisabetta Razzuoli

**Affiliations:** 1National Reference Center of Veterinary and Comparative Oncology (CEROVEC), Istituto Zooprofilattico Sperimentale del Piemonte, Liguria e Valle d’Aosta, Piazza Borgo Pila 39-24, 16129 Genova, Italy; chiaragrazia.deciucis@izsto.it (C.G.D.C.); livia.depaolis@izsto.it (L.D.P.); lisa.guardone@izsto.it (L.G.); elisabetta.razzuoli@izsto.it (E.R.); 2Department of Veterinary Medicine, University of Perugia, 06123 Perugia, Italy; katia.cappelli@unipg.it; 3Department of Animal Health, Istituto Zooprofilattico Sperimentale della Sardegna, 07100 Sassari, Italy; susanna.zinellu@izs-sardegna.it (S.Z.); giulia.franzoni@izs-sardegna.it (G.F.)

**Keywords:** goat EVs, RT-qPCR, IPEC-J2, Toll-like receptor, model of intestinal inflammation, anti-inflammatory, immunomodulating

## Abstract

Extracellular vesicles (EVs) are nanometric spherical structures, enclosed in a lipid bilayer membrane and secreted by multiple cell types under specific physiologic and pathologic conditions. Their complex cargo modulates immune cells within an inflammatory microenvironment. Milk is one of the most promising sources of EVs in terms of massive recovery, and milk extracellular vesicles (mEVs) have immunomodulatory and anti-inflammatory effects. The aim of this study was to characterize goat mEVs’ immunomodulating activities on Toll-like receptors (TLRs) and related immune genes, including cytokines, using a porcine intestinal epithelial cell line (IPEC-J2) after the establishment of a pro-inflammatory environment. IPEC-J2 was exposed for 2 h to pro-inflammatory stimuli as a model of inflammatory bowel disease (IBD), namely LPS for Crohn’s disease (CD) and H_2_O_2_ for ulcerative colitis (UC); then, cells were treated with goat mEVs for 48 h. RT-qPCR and ELISA data showed that cell exposure to LPS or H_2_O_2_ caused a pro-inflammatory response, with increased gene expression of *CXCL8*, *TNFA*, *NOS2* and the release of pro-inflammatory cytokines. In the LPS model, the treatment with mEVs after LPS determined the down-regulation of *NOS2*, *MMP9*, *TLR5*, *TGFB1*, *IFNB*, *IL18* and *IL12A* gene expressions, as well as lower release of IL-18 in culture supernatants. At the same time, we observed the increased expression of *TLR1*, *TLR2*, *TLR8* and *EBI3*. On the contrary, the treatment with mEVs after H_2_O_2_ exposure, the model of UC, determined the increased expression of *MMP9* alongside the decrease in *TGFB1*, *TLR8* and *DEFB1*, with a lower release of IL-1Ra in culture supernatants. Overall, our data showed that a 48 h treatment with mEVs after a pro-inflammatory stimulus significantly modulated the expression of several TLRs and cytokines in swine intestinal cells, in association with a decreased inflammation. These results further highlight the immunomodulatory potential of these nanosized structures and suggest their potential application in vivo.

## 1. Introduction

EVs are round structures of micro and nano-sized dimensions, surrounded by a phospholipid double-layer membrane [1]. They can be produced by all cell types, and they are released in the extracellular environment where they can be taken up by close receiving cells, or they may reach distant body sites through biological fluids [2,3]. Once absorbed by receiving cells, EVs can induce the modulation of biological processes through the release of the enclosed molecular cargo, the composition of which depends on the cell of origin and its patho/physiological conditions [4,5]. The main known functions for EVs are associated with the modulation of inflammation and immune-related molecules, as well as angiogenetic and pro-regenerative processes, depending on the cell of origin and patho/physiological conditions [5].

Milk EVs (mEVs) have recently gained attention for the promising application in the target therapy field, used as a drug delivery system [6,7,8]. This has been driven by the advantages that EVs have shown compared to synthetic therapeutic nanocarriers such as liposomes, revealing a wider biodistribution and bio-compatibility and a higher internalization rate [9]. Furthermore, milk allows to obtain a high number of vesicles, being a widely available and inexpensive raw material particularly enriched in EVs. Such mEV characteristics are ideal for theranostic applications and can be combined with their intrinsic immunomodulant characteristics, as shown in our previous studies [10,11,12,13].

Among immune related genes, Toll-like receptors (TLRs) are important molecules belonging to the family of pattern recognition receptors (PRRs). Recent studies have demonstrated that the dysregulation of TLRs may be associated with different inflammatory diseases, including intestinal bowel disease (IBD) [14,15,16,17]. TLRs help to recognize the self and non-self antigens, and play a pivotal role in the innate and adaptive immune responses, regulation of cytokine production, proliferation, and survival of the host cell. Moreover, they are directly involved in the regulation of inflammatory reactions for the elimination of infectious pathogens and cancer debris [18]. After recognition of the corresponding ligand, TLRs recruit toll-interleukin-1 receptor (TIR) domain-containing adaptor proteins such as MyD88, which subsequently activates NF-κB signaling and mitogen-activated protein kinases (MAPKs), resulting in the induction and release of pro-inflammatory cytokines. In addition, TLRs can also activate interferon regulatory factors (e.g., IRF3 and IRF7), triggering release of type I interferons [19]. To date, twenty-eight TLRs have been characterized in vertebrates. The highest number was described in fish, with twenty-one TLRs [20], while in mammals, thirteen TLRs have been identified. For some of these, such as TLR1-9 and TLR11 pathways, functions are known, while the roles of TLR10, 12 and 13 remain unclear. Humans and swine have ten TLRs [21], characterized by different localizations: TLR1, TLR2, TLR4, TLR5, TLR6, and TLR10 are located on the cell surface, whereas TLR3, TLR7, TLR8, TLR9, TLR11, TLR12, and TLR13 in the endosome [22]. Each TLR can recognize different self-compounds (e.g., ATP, HMGB1, cellular DNA or RNA) and not-self molecules (e.g., Flagellin, LPS, viral RNA or DNA), giving rise to the induction of inflammatory cytokine gene expression through the activation of NF-κB and MAPKs [21,23].

Inflammatory bowel disease (IBD) is a global and complex chronic disorder characterized by relapsing inflammatory events, accompanied by cell death and regeneration of the colon mucosa [24]. This phenomenon of alternate periods of damage and repair enhances the risk of neoplastic transformation within the cells of the intestinal epithelium [25]. IBD pathogenesis is multifactorial, involving genetic predisposition, mucosal barrier dysfunction, gastrointestinal microbiota disorders, immune response dysregulation, and environmental and lifestyle factors [26]. Aside from conventional therapies, novel drugs against IBD aim not only to induce and maintain symptom remission, but also to achieve mucosal healing by the elimination of local mucosal inflammation and the restoration of normal mucosal structure, as recently reviewed by Cai and co-workers [27]. Crohn’s disease (CD) and ulcerative colitis (UC) are two forms of IBD, with very different pathogenetic mechanisms and responses to therapeutic treatment. The intestinal cell inflammatory processes can be reproduced in vitro thanks to different pro-inflammatory stimulations [25,28]. In our previous studies, we reported anti-inflammatory and immune regulatory properties of both cow and goat milk extracellular vesicles (mEVs) on in vitro IBD models [12,13] focusing only on CD. Based on these previous results, and on the evidence that the effect of immunomodulatory molecules is often dose-dependent [29,30,31], in this study we aimed to further characterize the in vitro immunomodulatory effects of goat mEVs using IPEC-J2 (porcine jejunal epithelial cells). These intestinal porcine enterocytes were isolated from the jejunum of a suckling piglet. IPEC-J2 cells are neither transformed nor tumorigenic in nature, and express and produce cytokines, defensins, TLRs, and mucins [32]. These cells mimic the human physiology more closely than any other cell line. Therefore, they represent an ideal tool to study effects of probiotics, nutrients, and other compounds on a variety of parameters (e.g., transepithelial electrical resistance (TEER), permeability, metabolic activity), reflecting epithelial functionality [32] even for the colonic mucosa and being used to investigate pathogenetic mechanisms related to the colonic tract [33,34].

To analyze the innate defenses of inflamed swine intestinal cells, we focused on TLRs and related key immune genes, along with the release of several cytokines, using a one hundred-times lower dosage of mEVs with respect to the previous studies [12,13]. Moreover, here both IBD types were mimed (CD through LPS treatment and UC through H_2_O_2_), while in the past investigations, only LPS was used as pro-inflammatory stimulus. The two models of IBD here reported are based on the known role of LPS and H_2_O_2_ in these intestinal disorders, CD and UC, respectively. Indeed, CD is often associated with increased levels of LPS in the serum of suffering patients [28], while in UC, a causal role of H_2_O_2_ has been identified in both the pathogenesis and relapse of this IBD, and it is also a novel therapeutic target [35].

This is relevant as, although these two forms of IBD share similar clinical and pathological features, marked differences in the clinical presentation and response to treatments exist.

## 2. Results

Briefly, IPEC-J2 cells were treated for 2 h with LPS to mimic CD [36], or with H_2_O_2_ for UC simulation [25]. LPS is widely recognized as an activator of TLR4 and, consequently, of inflammation mediated by the activation of the MYD88/NF-Kb pathway [37]. Regarding H_2_O_2_, it determines NLRP3 inflammasome, and thereby mediates inflammation by IL-1β production and release [38]. After this step, mEVs (0.6 μg protein weight, which did not demonstrate toxic effects, see Appendix A) were added to the cell cultures for 48 h, the time at which cell gene expression and cytokine release in the supernatants were measured. As expected, treatment with either LPS or H_2_O_2_ caused a pro-inflammatory response: increased levels of pro-inflammatory cytokines (IL-6 and IL-8) were observed in culture supernatants of cells exposed to either LPS or H_2_O_2_ compared to controls (Appendix A).

### 2.1. Goat mEVs Effects on IPEC-J2 Gene Expression in the LPS Model

LPS treatment for 48 h determined in IPEC-J2 cells the increased gene expression of *CXCL8* (*p* < 0.0001) and *NOS2* (*p* = 0.0013), and the down-regulation of *TLR2* (*p* = 0.03), *NFKB1* (*p* = 0.005), *EBI3* (*p* = 0.00398), and *IFNB* (*p* = 0.039) (Figure 1, Figure 2 and Figure 3).

Concerning the IPEC-J2 exposure to mEVs after LPS stimulation on TLR family, an up-regulation of the gene expression of *TLR1* (*p* = 0.0346), *TLR2* (*p* = 0.0003), and *TLR8* (*p* = 0.0111), accompanied by a down-regulation of *TLR5* (*p* < 0.0001), and *MYD88* (*p* = 0.0042) was shown (Figure 1).

Regarding the cytokine and chemokine gene expression modulation, the mEV treatment after LPS stimulation induced an increase in *EBI3* (*p* < 0.0001), and a decrease in *IL12A* (*p* < 0.0001), *IL18* (*p* = 0.02), *TGFB1* (*p* < 0.0001), and *IFNB* (*p* < 0.0001) (Figure 2).

Moreover, mEV administration restored *NOS2* gene expression (*p* = 0.010) to the control levels after its increase induced by LPS treatment (Figure 3). Furthermore, the exposure to mEVs, following LPS treatment, down-regulated the expression of *DEFB1* (*p* = 0.0069) and *MMP9* (*p* = 0.0389).

### 2.2. Goat mEV Effects on IPEC-J2 Gene Expression in the H_2_O_2_ Model

The H_2_O_2_ treatment determined the up-regulation of *TLR2* (*p* = 0.0017), TLR8 (*p* = 0.0002), *NFKB1* (*p* = 0.0011) and *TGFB1* (*p* = 0.0018), accompanied by the down-regulation of *TLR5* (*p* = 0.0012), *IL6* (*p* = 0.014), IFNA (*p* = 0.0021), *IFNB* (*p* = 0.0015), and *MUC2* (*p* < 0.0001) (Figure 4, Figure 5 and Figure 6). mEV administration after H_2_O_2_ treatment induced the increased gene expression of *TLR7* (*p* = 0.0096) and the down-regulation of *TLR8* (*p* = 0.012), which returned close to the basal level of the control (Figure 4).

Concerning cytokines, mEV exposure determined the gene expression decrease in *IL18* (*p* = 0.0099) and *TGFB1* (*p* = 0.0091) in IPEC-J2 cells pretreated with H_2_O_2_ (Figure 5).

Furthermore, the exposure to mEVs down-regulated the expression of *DEFB1* (*p* = 0.021). Interestingly, an up-regulation of *MUC2* (*p* = 0.032) and *MMP9* (*p* = 0.0389) was found after the mEV treatment.

### 2.3. Cytokine Quantification

To further investigate the immunomodulatory properties of goat mEVs in an inflammatory environment, we measured the cytokine contents in the supernatants of IPEC-J2 stimulated with LPS in absence or presence of mEVs (0.6 μg protein weight), 48 h post-stimulation (Figure 7). As expected, LPS stimulation resulted in enhanced release of pro-inflammatory cytokines, such as IL-1α (*p* = 0.0012), IL-6 (*p* = 0.0387), IL-8 (*p* = 0.0028), and IL-18 (*p* = 0.0008), in accordance with our recent study [12]. The administration of mEVs after LPS did not alter the release of all the tested cytokines compared to LPS stimulation alone, except for a small reduction in IL-1α and IL-18, although without statistical significance (Figure 7).

The cytokine release in the IPEC-J2 culture supernatants was also measured in the H_2_O_2_ experiment, mimicking UC (Figure 8). The H_2_O_2_ treatment resulted in the increased release of pro-inflammatory IL-6 (*p* = 0.0305) and IL-8 (*p* = 0.0257), as well as in a milder increase in IL-1α and of the anti-inflammatory IL-1Ra, although without statistical significance (Figure 8).

The administration of mEVs did not impact culture supernatant levels of all the tested cytokines, including failure to reduce IL-6 and IL-8, which were increased by H_2_O_2_ treatment. Only IL-1Ra was reduced by mEV administration (*p* = 0.0155) (Figure 8).

## 3. Discussion

In recent years, mEVs have increasingly attracted interest due to their immunomodulatory and anti-inflammatory potential [10,11]. IBD is a multifactor complex chronic disorder characterized by relapsing inflammatory events accompanied by cell death and regeneration of the colon mucosa [27,39,40,41]. Other authors already emphasized the possibility to use EVs for their anti-inflammatory properties, thanks to their miRNA cargo that may regulate the gene expression of recipient cells [27,42], even as therapeutic tools against intestinal disorders [17]. Recently, our research group also has focused on mEVs, testing their efficacy on different in vitro IBD models, demonstrating cow and goat mEVs anti-inflammatory activities [10]. However, in our previous papers, only LPS was used to mimic CD and establish the pro-inflammatory environment where mEVs were tested. However, the two forms of IBD, i.e., Crohn’s disease (CD) and ulcerative colitis (UC), are very different for pathogenetic mechanisms and therapeutic treatment responses [25,36]. Thus, in this study, we deeply investigated the molecular mechanisms underlying the anti-inflammatory effect of goat mEVs, applying two models that mimic CD (using LPS) [36] and UC (using H_2_O_2_) [25]. The effect of goat mEVs on the expression of immune genes were investigated, in particular on TLRs and on the production of molecules involved in innate immune system for antigen recognition and immune response activation. Moreover, based on previously obtained data [12] and on the literature related to the dose-dependent effects of immunomodulant agents [29,30,31], we used a concentration of mEVs one hundred times lower than the concentration used in our previous paper [12], in order to evaluate a possible different effect related to the dosage.

The first result, that confirms our hypothesis on mEV immunomodulatory effects in relation to the dosage, is the expression of the *CXCL8* gene (encoding IL8) and the cytokine release. Indeed, although *CXCL8* gene expression increased after inflammatory stimuli, it did not further increase after the administration of mEVs, as observed in the previous paper with a hundred higher dose of vesicles [12]. As IL8 is pro-inflammatory, its control is essential in IBD therapy. Similarly, the release of other proinflammatory cytokines (such as IL1α and IL6) is comparable to those of the LPS stimulus alone, while significantly increase if mEVs are used in a one-hundred times higher concentration [12]. On the contrary, a lower release of IL18 after mEVs + LPS treatment compared to LPS alone was observed.

Regarding the effect of mEVs on the expression of TLR family genes, in the LPS model we observed an up-regulation of *TLR1*, *2* and *8*, as well as the down-regulation of *TLR5.* With respect to previous results, a lower mEV concentration seems to fail to modulate *TLR4* and *TLR7* in the LPS model [12]. In the H_2_O_2_ model, *TLR8* was down-regulated and *TLR7* was up-regulated by mEV treatment.

The Increase in the production of *TLR1* observed after mEV treatment in the LPS model may be protective against chronic inflammation, since the TLR1 pathway is crucial in the immune response against Gram-negative pathogens [43]. The absence of TLR1 expression during an acute infection causes chronic immune activation and the transmutation of microbiota [44]. This background indicates that TLR1 pathway activation may prevent the colon chronic inflammation during IBD. On the other hand, TLR2 possesses the ability to reinforce the tight-junctions in epithelial cells (survival and proliferation signals), and to induce tolerogenic responses through dendritic cells (polarizing the immune response towards a T regulator phenotype) [45]. Even if no changes have been detected in TLR2 expression levels in UC and CD patients [46], the up-regulation induced by mEV treatment in our LPS model may be helpful for the attenuation of symptoms.

mEV administration in the LPS model also reduced the expression of *TLR5*, which is the receptor for bacterial flagellin, known as a key factor in triggering the inflammatory status in CD [47]. Indeed, the interaction between TLR5 and flagellin determines an inflammatory response by the intestinal epithelia. Human and murine studies showed its importance in the regulation of innate and adaptive immune responses associated with IBD [47]. Moreover, TLR5 activation in an ex vivo model of IBD determined a decrease in epithelial barrier resistance and the altered expression of tight junction proteins (claudin-3, occludin and zonula occludens-1) compared with controls [47]. These data suggest that elevated expression of *TLR5*, resulting in the inability to maintain barrier function in response to bacterial flagellin, can lead to CD-like ileitis susceptibility [47]. In this contest, mEV treatment may have a protective effect due to the down-regulation induced on this receptor.

The involvement in IBD pathogenesis of TLR7 and TLR8 receptors is not well known to date. In particular, Fernandes and co-workers [46] demonstrated that TLR8 expression did not significantly differ in active and inactive CD compared to healthy subjects [46]. A different consideration can be made for UC, where TLR7/8 are also involved in many intracellular pathways culminating with the expression of pro-inflammatory molecules. It is known that TLR8 activation can enhance TNF and IL-1β production, both associated with mucosal inflammation in UC [48], whereas TLR7 agonists could induce a type I IFN response and prevent experimental colitis in mice [48]. For these reasons, we can speculate that mEV treatment, which down-regulated these receptors, may reduce these pathways, with a beneficial effect on UC pathology [49].

Moreover, mEV treatment induced, in both models, the down-regulation of the *DEFB1* gene encoding for an antimicrobial peptide produced by a variety of epithelial cells. Indeed, within the host immune response, beta-defensin peptides are involved in protection and tolerance balance between pathogen and non-pathogen flora. Although in our previous study [12] a hundred-times higher concentration of mEVs was able to up-regulate the gene expression of *DEFB1* in the LPS model, here, with a lower dosage, we observed a decreased expression. This is a particularly interesting finding, since this peptide was found to increase in UC patients [50], and also confirms what is quite well known for immunomodulant substances which may induce opposite effects based on their concentration [29,30,31].

Moreover, to date, an increased production in CD mucosa of IL-12, the major Th1-inducing cytokine in human, is well known [51]. IL-12, together with IL-23, IL-27 and IL-35, belongs to the interleukin 12 family. These cytokines consist of an α chain (p19, p28 or p35) and a β chain (p40 or Ebi3). The p40 (*IL12B*) can pair with p35 (*IL12A*) or p19 to form IL-12 or IL-23, respectively. Ebi3 can pair with p28 or p35 to form IL-27 or IL-35, respectively [52]. IL-12 and IL-23 are predominantly pro-inflammatory/pro-stimulatory cytokines, and are involved in the development of Th1 and Th17 cells [53]. IL-27 is an immunoregulatory cytokine and IL-35 is a potent inhibitory interleukin. This establishes a functional balance within this family, with IL-12 and IL-23 as positive regulators, and IL-27 and IL-35 as negative regulators [44]. In this study, we detected the down-regulation of the *IL12A* subunit and the up-regulation of the *EBI3* subunit following mEV administration in the LPS model, suggesting a reduction in IL-12 and an increase in IL-27. In our previous study [12], with a hundred times higher administration of mEVs, we showed an up-regulation of both *IL12A* and *EBI3* subunit and, therefore, a potential increase in IL35, as previously mentioned. However, both IL-35 and IL-27 are negative regulators, indicating a possible similar effect through different pathways. Interestingly, these results were associated with the down-regulation of IL-18 expression and secretion in the LPS model of both studies [54].

IL-18 is a member of the IL-1 superfamily, and a potent inducer of IFN-γ production, which plays a crucial role in the T helper cell type 1 (Th1) response during immunorecognition [55]. It can also enhance other T-cell responses, such as Th17 in synergy with IL-23, or Th2 in the absence of IL-12, IL-15, or IL-23 [56]. Although IL-18 has a protective role in the early phase of inflammation [57], increased levels of this cytokine were observed in intestinal pathological conditions, such as Crohn’s disease [57]. IL-18 release can be damaging, since it can enhance leukocyte recruitment and promote severe inflammation, with subsequent dysbiosis [57]. Our data revealed that goat mEVs can reduce the production of IL-18 in an inflammatory condition (both H_2_O_2_ and LPS treatments). In the LPS model a combination between IL-12 and IL-18 production in the Th1 cell differentiation was shown: IL-12 is sufficient to trigger the activation of these cells through the induced synthesis of IFN-γ, while IL-18 has a pivotal role in perpetuating Th1 cell response [58]. Moreover, functional studies showed that the down-regulation of IL-18 expression in cultures of CD lamina propria mononuclear cells, by specific IL-18 antisense oligonucleotides, significantly inhibited IFN-γ synthesis, further supporting the concept that IL-18 serves as a strong co-stimulatory factor of IL-12-driven Th1-activation [59,60]. In this study, we demonstrated the possible down-regulation of this axis by the mEV treatment in an IBD model.

Finally, the IL-1Ra is another member of the IL-1 superfamily and a receptor antagonists [61]. IL-1Ra binds the receptor IL-1R1 with a higher affinity than IL-1α or IL-1β, although without activation of the IL-1 signaling [61]. Indeed, this cytokine is released to block IL-1 activity, preventing the development of an exacerbated inflammatory immune response [61]. Both IL-1α and IL-1β have a protective role in the early phase of inflammation, and it was described that IL-1β promoted Th17 responses and sustained both innate and adaptive inflammatory responses in the gut in synergy with IL-23 [62]. Its receptor antagonist IL-1Ra regulates normal immune homeostasis in the gut [62]. Both IL-1 and IL-1Ra are up-regulated in IBD [62]. Our data revealed that goat mEVs reduced the production of IL-1Ra in an inflammatory condition (H_2_O_2_ treatment) which, in vivo, might enhance IL-1β activity in UC conditions.

## 4. Materials and Methods

### 4.1. Milk Collection

Goat milk was collected from one local farm in Umbria (Italy), previously selected and monitored by the Department of Veterinary Medicine (University of Perugia). The samples were obtained from bulk tank milk, in order to avoid inter-individual variability, in the late summer/autumn period. Before processing, milk was stored for less than 24 h at 4 °C, avoiding any intermediate cryo-preservation in order to preserve vesicle morphology and reduce artefacts.

### 4.2. Extracellular Vesicle (EV) Isolation, Characterization and Size Distribution Assessment

mEVs were isolated as previously described by Mecocci and collaborators [12,13]. Briefly, serial differential centrifugations, alternated with a step in ethylenediaminetetraacetic acid tetrasodium salt dihydrate (EDTA; cod. 11836170001, Merck Life Science S.r.l., Milan, Italy), and a final ultracentrifugation to recover mEVs in the pellet, were carried out. Total protein concentration was measured using Bradford assay (cod. 5000201 Bio-Rad, Milan, Italy).

Moreover, mEV isolation was verified through morphological characterization with a transmission electron microscopy (TEM), nanoparticle tracking assay (NTA-Malvern, Worcestershire, UK), and ExoView^TM^ R100 technology (NanoView Biosciences, Brighton, MA, USA), as reported in [12,13], where the same batch of mEVs were utilized.

### 4.3. Cell Cultures

IPEC-J2 cells (porcine jejunal epithelial cells, IZSLER Cell Bank code BS CL 205) were cultured in a complete medium consisting of a mixture (1:1) of Dulbecco’s modified Eagle high glucose medium (DMEM, cod. ECM0101L, Euroclone, Milan, Italy) and nutrient mixture F-12 (F12, cod. ECB7502L, Euroclone, Milan, Italy) enriched with 10% fetal bovine serum (FBS, GIBCO™, cod. A38401, Thermofisher scientific, Milan, Italy), 1% L-glutamine 2 mM solution (cod. ECB3000D, Euroclone, Milan, Italy) and 1% penicillin/streptomycin solution (cod. ECB3001D, Euroclone, Milan, Italy). Cells were seeded into 12 well plates (1 mL per well, 3 × 10^5^ cells/mL; Euroclone, Milan, Italy) and then incubated at 37 °C, 5% CO_2_, until confluence.

#### IPEC-J2 Models of Inflammatory Bowel Disease to Mimic Crohn’s Disease and Ulcerative Colitis

To mimic CD, named the “LPS model” here in the text, IPEC-J2 was treated with purified lipopolysaccharides (LPS) (1 μg/mL; from *Escherichia coli* 0111:B4, cod. L5293, Merck KGaA, Darmstadt, Germany) for 2 h [36,39,40]. Then, cells were washed with PBS (cod. 524650-1EA, Merck Life Science S.r.l., Milan, Italy) and treated with a 0.6 μg protein weight of mEVs suspension or only with complete medium, and incubated at 37 °C, 5% CO_2_ for 48 h. Concerning the simulation of UC, named the “H_2_O_2_ model” here in the text [41,63,64], IPEC-J2 was treated with 200 μM H_2_O_2_ for 2 h; after this, cells were washed and exposed to 0.6 μg protein weight mEVs or only to complete medium and incubated at 37 °C, 5% CO_2_ for 48 h. Untreated cells (only exposed to medium) were used as controls for each experiment and incubated in the same condition of treated cells. The choice of mEV concentration to be administered to the cells is derived from previous cell viability experiments performed in our previous studies [12,13], as well as the choice of the LPS concentration. As far as the concentration of H_2_O_2_ is concerned, it is based on previously published papers by other authors [64,65]. At the end of each time point cells were harvested and lysed with 400 μL of RLT buffer (Qiagen, cod. 79216, Hilden, Germany) and, after incubation for 10 min at room temperature (RT), they were collected and stored at −80 °C until use. The experiment was repeated three times, with three technical replicates in each experiment, using mEV isolates derived from different goat milk samples. Summarizing, cell gene expression and cytokine release in the supernatants were assessed in untreated cells (Control), in cells inflamed with LPS (thereafter named “LPS”) or H_2_O_2_ (“H_2_O_2_”) and in inflamed cells treated with mEVs (“LPS + mEVs”; “H_2_O_2_ + mEVs”). Supernatants were collected forty-eight (48) h post-treatment and stored at −80 °C until multiplex ELISA assay.

### 4.4. XTT Assay

We tested different mEV concentrations in terms of protein weight (0.006 μg, 0.6 μg, 60 μg). A 2,3-bis-(2-methoxy-4-nitro-5-sulfophenyl)-2H-tetrazolium-5-carboxanilide (XTT) assay was performed according to the manufacturer’s instructions (cod., X12223 XTT Cell Viability Kit, Cell Signaling Technology Inc., Danvers, MA, USA). In brief, IPEC-J2 cells were plated on a 96-well plate and incubated for 24 h at 37 °C until confluence. The day after, cells were exposed to mEVs and then incubated again for 48 h at 37 °C, 5% CO_2_. Untreated cells were used as controls. The absorbance was measured at 450 nm using a multimode microplate reader (Glomax^®^, Promega™, Milan, Italy). This assay was performed three times for each concentration, setting up four technical replicas each.

### 4.5. RNA Extraction and RT-qPCR

Total RNA was extracted from IPEC-J2 cells, as previously indicated, using RNeasy Mini Kit (Qiagen s.r.l., cod. 74004, Milan, Italy) through the Qiacube System (Qiagen s.r.l., Milan, Italy) in accordance with the manufacturer’s instructions. RNA extraction was assessed using Qubit 3.0 Fluorometer (ThermoFisher Scientific, Waltham, MA, USA). For each sample, 250 ng of RNA was reverse-transcribed into cDNA using an iScript^®^ cDNA Synthesis Kit (Bio-Rad, cod. 1708891, Milan, Italy). Amplification was performed on a CFX96™ Real-Time System (Bio-Rad, Milan, Italy) using the SsoFast™ EvaGreen^®^ Supermix (Bio-Rad, cod., 1725200 Milan, Italy), following a protocol previously described [12,66]. In this study, we tested the expression of *IL12A*, *IL12B*, Epstein–Barr virus-induced gene 3 (*EBI3)*, interleukin 6 (*IL6*), IL-8 coding gene C-X-C motif chemokine ligand 8 (*CXCL8*), IL18, tumor necrosis factor alpha (*TNFA*), transforming growth factor beta 1 (*TGFβ1*), nitric oxide synthase 2 (*NOS2*), mucin 2 (*MUC2),* matrix metallopeptidase 9 (*MMP9*), Toll-like receptor 1 (*TLR1*), *TLR2*, *TLR3*, *TLR4*, TLR5, *TLR7*, *TLR8, TLR9,* defensina beta (DEFB1) *1, DEFB4A,* mieloid differentiation primary response 88 (*MYD88),* nuclear factor kappa B subunit 1 (*NFKB1),* NFKB-p65 subunit (*RELA*)*,* interferon alfa 1 *(IFNA1),* interferon beta *(IFNB),* and interferon Regulatory Factor 3 (*IRF3),* while glyceraldehyde 3-phosphate dehydrogenase *(GAPDH)* was used as gene reference as previously reported [12]. The primer set is described in Appendix A.

### 4.6. RT-qPCR Analyses

A normalization step was performed according to the expression levels of the reference genes after assessing their stability under different experimental conditions, using the norm algorithm included in Bio-Rad CFX Maestro software (ver. 4.1 BioRad, Hercules, CA, USA). Relative normalized expression was assessed using the 2^−∆∆CT^ method [67] comparing different conditions (“Control” vs. “LPS” for 48 h, “LPS + mEVs ” vs. “LPS” for 48 h, “Control” vs. “H_2_O_2_” for 48 h, “H_2_O_2_ + mEVs vs. H_2_O_2_” for 48 h) in the cell cultured model. The data were analyzed using GraphPad Prism 5.04 (GraphPad Software Inc., La Jolla, CA, USA). Gene expression data were submitted to a Kolmogorov–Smirnov test to check Gaussian distributions. Significant differences were checked by Kruskal–Wallis test and applying the post hoc Dunn’s multiple comparison test. The significance threshold was set at *p* < 0.05.

### 4.7. Cytokine Quantification

mEVs’ impact on cytokine production by IPEC-J2 was investigated using multiplex ELISA, which allows simultaneous quantification of different cytokines. In detail, culture supernatants were collected 48 h post-stimulation, centrifuged (at 2500× *g* for 3 min) and kept at −80 °C until analyzed. Levels of IL-1α, IL-1β, IL-1Ra, IL-6, IL-8, IL-10, IL-12, IL-18, and GM-CSF were determined using the Porcine Cytokine/Chemokine Magnetic Bead Panel Multiplex Assay (Merck Millipore, cod. PCYTMG-23K-13PX, Darmstadt, Germany) and a Bioplex MAGPIX Multiplex Reader (Bio-Rad, Hercules, CA, USA), following the manufacturer’s recommendations [12]. The data were analyzed using GraphPad Prism 9.01 (GraphPad Software Inc., La Jolla, CA, USA).

### 4.8. Statistical Analyses

Data were submitted to a Kolmogorov–Smirnov test to check Gaussian distributions. Differences were evaluated through ANOVA followed by Dunnet’s multiple comparison test, or a Kruskal–Wallis test followed by Dunn’s multiple comparison test. *p*-value < 0.05 was considered statistically significant.

## 5. Conclusions

Dysfunction of the immune system, and of TLRs in particular, plays a key role in IBD pathogenesis. Although many therapies to treat IBD have been attempted, they often have side effects on the gastrointestinal tract, or are not fully effective [27]. Therefore, TLRs could be considered a new therapeutic target for IBD patients. Our study investigated the ability of mEVs to modulate the expression of TLRs for the first time, showing that the concentration used is associated with a modulation of these and other genes and proteins. Moreover, we speculate a possible immunomodulant action of these vesicles given by their molecular cargo, possibly in a dose-dependent manner.

## Figures and Tables

**Figure 1 ijms-24-11096-f001:**
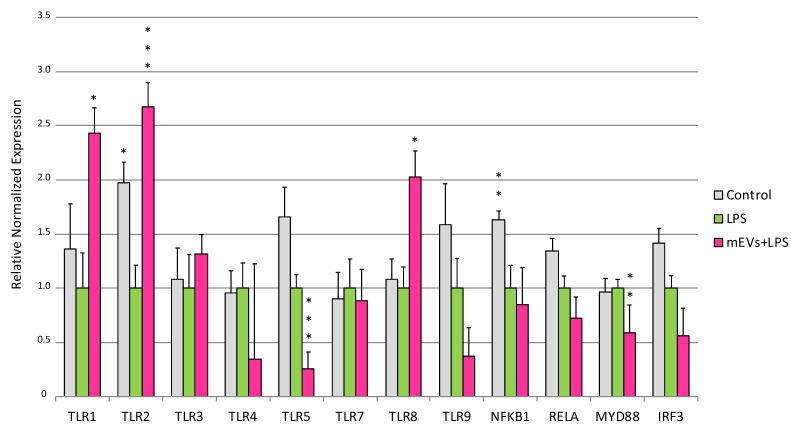
Goat mEVs’ effects on TLR family genes. Evaluation of IPEC-J2 gene expression after the exposure to LPS to mimic CD condition. IPEC-J2 cells were left untreated (control, grey), stimulated with LPS (inflamed condition, green) or treated with LPS and a 0.6 μg protein weight mEV suspension (fuchsia). Differences (Control/LPS + mEVs vs. LPS) were evaluated through Kruskal–Wallis test followed by Dunn’s multiple comparison test; * *p* < 0.05, ** *p* < 0.01 and *** *p* < 0.001.

**Figure 2 ijms-24-11096-f002:**
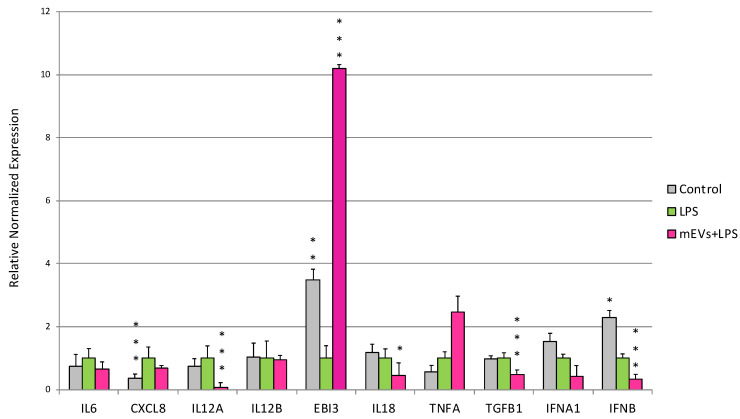
Goat mEVs’ effects on IPEC-J2 gene expressions after exposure to LPS to mimic CD condition. IPEC-J2 cells were left untreated (control, grey), stimulated with LPS (inflamed condition, green) or treated with LPS and a 0.6 μg protein weight mEV suspension (fuchsia). Differences (Control/LPS + mEVs vs. LPS) were evaluated through Kruskal–Wallis test followed by Dunn’s multiple comparison test; * *p* < 0.05, ** *p* < 0.01 and *** *p* < 0.001.

**Figure 3 ijms-24-11096-f003:**
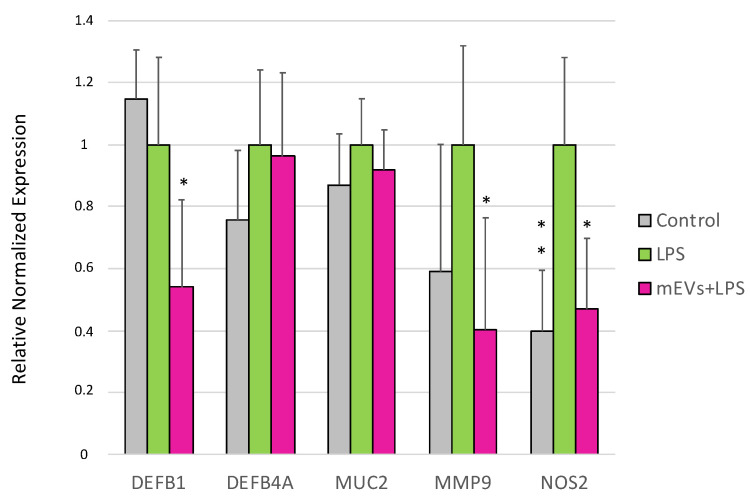
Goat mEVs’ effects on IPEC-J2 gene expressions after exposure to LPS to mimic CD condition. IPEC-J2 cells were left untreated (control, grey), stimulated with LPS (inflamed condition, green) or treated with LPS and 0.6 μg protein weight mEVs suspension (fuchsia). Differences (Control/LPS + mEVs *vs.* LPS) were evaluated through Kruskal–Wallis test followed by Dunn’s multiple comparison test; * *p* < 0.05 and ** *p* < 0.01.

**Figure 4 ijms-24-11096-f004:**
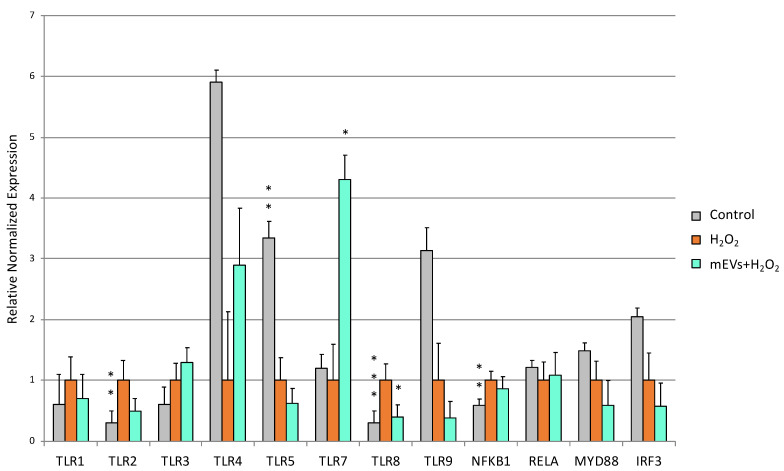
Goat mEVs’ effects on IPEC-J2 gene expressions of TLR family genes after exposure to H_2_O_2_ to mimic UC condition. IPEC-J2 cells were left untreated (control, grey), stimulated with H_2_O_2_ (inflamed condition, orange) or treated with H_2_O_2_ and 0.6 μg protein weight mEVs suspension (teal). Differences (Control/H_2_O_2_ + mEVs vs. H_2_O_2_) were evaluated through Kruskal–Wallis test followed by Dunn’s multiple comparison test; * *p* < 0.05, ** *p* < 0.01 and *** *p* < 0.001.

**Figure 5 ijms-24-11096-f005:**
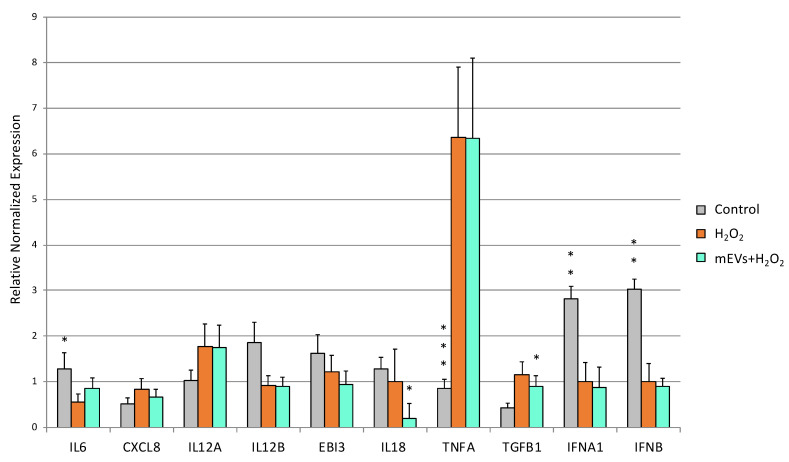
Goat mEVs’ effects on IPEC-J2 gene expressions after exposure to H_2_O_2_ to mimic UC condition. IPEC-J2 cells were left untreated (control, grey), stimulated with H_2_O_2_ (inflamed condition, orange) or treated with H_2_O_2_ and 0.6 μg protein weight mEVs suspension (teal). Differences (Control/H_2_O_2_ + mEVs vs. H_2_O_2_) were evaluated through. Kruskal–Wallis test followed by Dunn’s multiple comparison test; * *p* < 0.05, ** *p* < 0.01 and *** *p* < 0.001.

**Figure 6 ijms-24-11096-f006:**
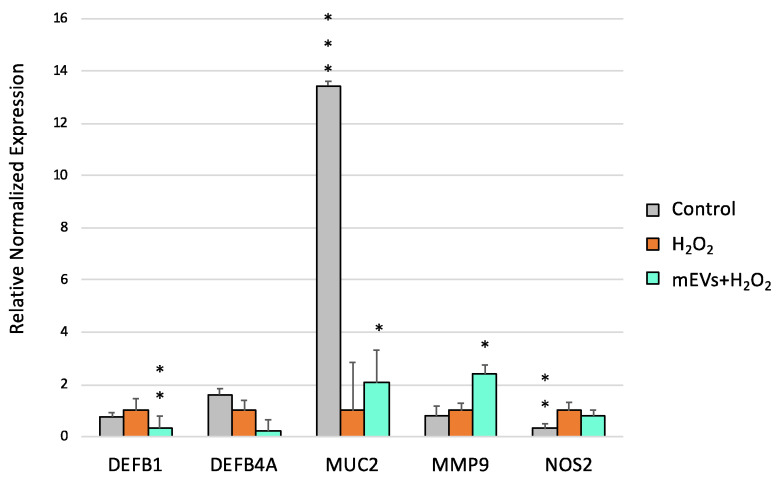
Goat mEVs’ effects on IPEC-J2 gene expressions after exposure to H_2_O_2_ to mimic UC condition. IPEC-J2 cells were left untreated (control, grey), stimulated with H_2_O_2_ (inflamed condition, orange) or treated with H_2_O_2_ and 0.6 μg protein weight mEVs suspension (teal). Differences (Control/H_2_O_2_ + mEVs vs. H_2_O_2_) were evaluated through Kruskal–Wallis test followed by Dunn’s multiple comparison test; * *p* < 0.05, ** *p* < 0.01 and *** *p* < 0.001.

**Figure 7 ijms-24-11096-f007:**
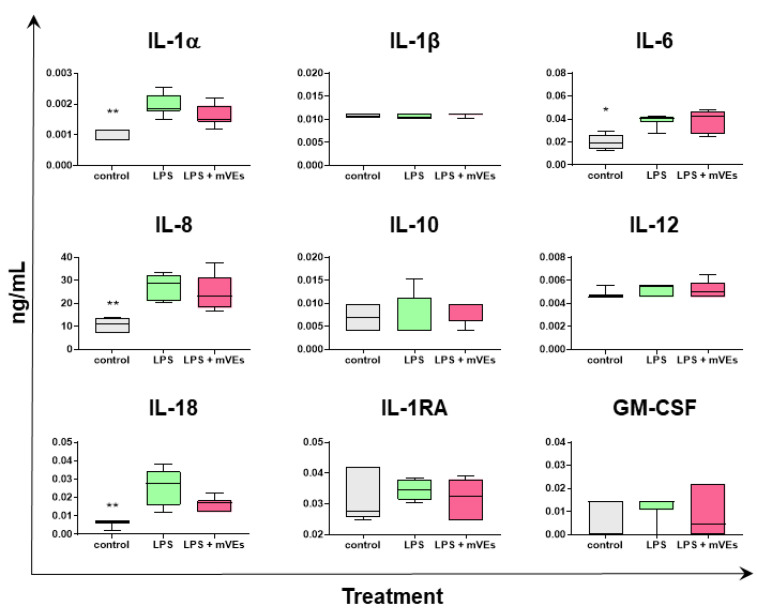
mEVs’ impact on cytokine production by IPEC-J2 in the LPS model. IPEC-J2 cells were left untreated (control, grey), stimulated with LPS (inflamed condition, green), or treated with LPS and 0.6 μg protein weight mEVs suspension (pink). A total of 48 h after the mEVs treatment, culture supernatants were collected, and levels of several cytokines were determined through multiplex ELISA. Data are presented as box and whisker plots displaying median and interquartile range (boxes) and minimum and maximum values (whiskers). Differences (control/LPS + mEVs vs. LPS) were evaluated through ANOVA followed by Dunnett’s multiple comparison test, or a Kruskal–Wallis test followed by Dunn’s multiple comparison test; * *p* < 0.05, ** *p* < 0.01.

**Figure 8 ijms-24-11096-f008:**
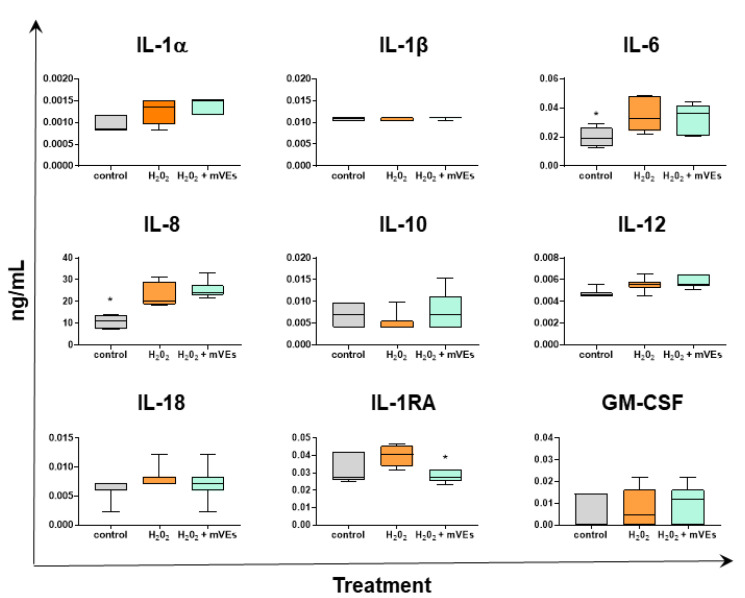
mEVs’ impacts on cytokine production by IPEC-J2 in the H_2_O_2_ model. IPEC-J2 cells were left untreated (control, grey), stimulated with H_2_O_2_ (inflamed condition, orange), or treated with H_2_O_2_ and 0.6 μg protein weight mEV suspension (teal). A total of 48 h after the mEVs treatment, culture supernatants were collected, and levels of several cytokines were determined through multiplex ELISA. Data are presented as box and whisker plots displaying median and interquartile range (boxes) and minimum and maximum values (whiskers). Data were submitted to a Kolmogorov–Smirnov test to check Gaussian distributions. Differences (control/H_2_O_2_ + mEVs vs. H_2_O_2_) were evaluated through ANOVA followed by Dunnett’s test, or a Kruskal–Wallis test followed by Dunn’s test; * *p* < 0.05.

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
