# Peer review of "Toll-like Receptors and Cytokine Modulation by Goat Milk Extracellular Vesicles in a Model of Intestinal Inflammation"

_ijms, 2023, doi:10.3390/ijms241311096_

Round 1

Reviewer 1 Report

In the current study, the authors investigated whether goat milk extracellular vesicles possess immunomodulating activities on toll like receptors and related immune genes, including cytokines, using a porcine intestinal epithelial cell line (IPEC-J2) after the establishment of a pro-inflammatory environment.

Some revisions need to be done before acceptance.

1)  General Revision:

-   Typography: the authors should read thoroughly their manuscript and check: 1) space between words; 2) English of some sentences

-   I suggest to add an abbreviation list; there are so many abbreviations in the manuscript and the readers are not familiar with them.

2) Introduction:

I suggest to add a small introduction on intestinal porcine epithelial cell line-J2 (IPEC-J2).

3) Material and Methods section:

-     Specify the code of all materials.

-     Please, justify the use of LPS (1 μg/mL), H2O2 (100 μM) and mEVs (0.6 μg protein weight).

-     I suggest to perform an MTT assay, to evaluate the effect of mEV on IPEC-J2 cell viability

-     I suggest to briefly explain the multiplex ELISA assay.

4)  Conclusions:

-  Do the authors consider the idea of also carrying out an in vivo study to confirm the potential of mEVs?

In the current study, the authors investigated whether goat milk extracellular vesicles possess immunomodulating activities on toll like receptors and related immune genes, including cytokines, using a porcine intestinal epithelial cell line (IPEC-J2) after the establishment of a pro-inflammatory environment.

Some revisions need to be done before acceptance.

1)  General Revision:

-   Typography: the authors should read thoroughly their manuscript and check: 1) space between words; 2) English of some sentences

-   I suggest to add an abbreviation list; there are so many abbreviations in the manuscript and the readers are not familiar with them.

2) Introduction:

I suggest to add a small introduction on intestinal porcine epithelial cell line-J2 (IPEC-J2).

3) Material and Methods section:

-     Specify the code of all materials.

-     Please, justify the use of LPS (1 μg/mL), H2O2 (100 μM) and mEVs (0.6 μg protein weight).

-     I suggest to perform an MTT assay, to evaluate the effect of mEV on IPEC-J2 cell viability

-     I suggest to briefly explain the multiplex ELISA assay.

4)  Conclusions:

-  Do the authors consider the idea of also carrying out an in vivo study to confirm the potential of mEVs?

Author Response

Thanks to the referee for the suggestions. We have revised the paper accordling

1)  General Revision:

-   Typography: the authors should read thoroughly their manuscript and check: 1) space between words; 2) English of some sentences

Thanks, the manuscript was read thoroughly and typos were corrected.

-   I suggest to add an abbreviation list; there are so many abbreviations in the manuscript and the readers are not familiar with them.

Abbreviation: EV: extracellular vesicles; mEV: milk extracellular vesicles; TLR: Toll Like Receptor; PRRs: pattern recognition receptors IBD: intestinal bowel disease; MyD88: Myeloid Differentiation Primary Response Protein 88, NF-κB: Nuclear Factor Kappa B; MAPKs: mitogen-activated protein kinases; IRF interferon regulatory factor; ATP: adenosine triphosphate; HMGB1: High Mobility Group 1; DNA ; deoxyribonucleic acid; RNA: ribonucleic acid; LPS: lipopolysaccharide; CD: Crohn Disease; UC: Ulcerative Colitis; TEER: transepithelial electrical resistance; NLRP3: nucleotide-binding oligomerization domain, leucine rich repeat and pyrin domain containing 3; IL: interleukin; EBI3: Epstein-Barr virus induced gene 3; CXCL8: IL-8 coding gene C-X-C Motif Chemokine Ligand 8; TNFA: Tumor Necrosis Factor alpha; TGFB1: Transforming Growth Factor Beta 1; NOS2: Nitric Oxide Synthase 2; MUC2: Mucin 2; MMP9: Matrix metallopeptidase 9; DEFB: defensina beta; RELA NFKB-p65 subunit; IFN: Interferon; GAPDH: Glyceraldehyde 3-phosphate dehydrogenase; Th1: T helper cell type 1; IL1Ra: Interleukin 1 Receptor Antagonist; EDTA: ethylenediaminetetraacetic acid tetrasodium salt dehydrate; TEM: Transmission Electron Microscopy; NTA: Nanoparticle Tracking Assay; DMEM: Dulbecco's Modified Eagle High Glucose; F12: Nutrient Mixture F-12; FBS: Fetal Bovine Serum; RLT: lysis buffer for RNA isolation; RT: Room Temperature; ELISA: enzyme-linked immunosorbent assay; XTT: 2,3-bis-(2-methoxy-4-nitro-5-sulfophenyl)-2H-tetrazolium-5-carboxanilide; PBS: phosphate buffer saline; RT-qPCR: real time quantitative polymerase chain reaction; cDNA: complementaryDNA; GM-CSF: Granulocyte-Macrophage Colony-Stimulating Factor

2) Introduction:

I suggest to add a small introduction on intestinal porcine epithelial cell line-J2 (IPEC-J2).

Reply. A small introduction was added on using IPEC-J2(porcine jejunal epithelial cells). These intestinal porcine enterocytes were isolated from the jejunum of a suckling piglet. IPEC-J2 are unique as derived from the small intestine and neither transformed nor tumorigenic in nature. These cells mimic the human physiology more closely than any other cell line. Therefore, they represent an ideal tool to study effects of probiotic, nutrients and other compounds on a variety of parameters (e.g. transepithelial electrical resistance (TEER), permeability, metabolic activity) reflecting epithelial functionality (DOI:10.1007/978-3-319-16104-4_12)

3) Material and Methods section:

-     Specify the code of all materials.

We have added such information

-Please, justify the use of LPS (1 μg/mL), H2O2 (100 μM) and mEVs (0.6 μg protein weight).

Reply: The choice of mEVs concentration to be administered to the cells is derived from previous cell viability experiments performed in our previous papers, by Mecocci and collaborators (Reference n.12 and n.13), as well as the choice of the LPS concentration. As far as the concentration of H2O2 is concerned, it is based on previously published papers (https://doi.org/10.3390/ani13081401; https://doi.org/10.1186/s10020-020-00165-3) and it was of 200 μM. Citations and this explanation were added to the text.

-     I suggest to perform an MTT assay, to evaluate the effect of mEV on IPEC-J2 cell viability

Reply:The effect of goat mEV in IPEC-J2 was added. We decided to use an XTT test because it is routinely performed by our group. Concerning the concentration, we have chosen three different concentrations: high-dose (60 µg protein weight), medium-dose (0.6 µg protein weight) and low-dose to test cytotoxicity (0.006 µg protein weight). A paragraph was added in the text.

-     I suggest to briefly explain the multiplex ELISA assay.

Reply: Multiplex ELISA is an immunoassay which allows simultaneous quantification of diverse cytokines. This information was added to the text in Section 4.6.

4)  Conclusions:

-  Do the authors consider the idea of also carrying out an in vivo study to confirm the potential of mEVs?

Reply: Thanks to the referee for the suggestion. Of course, we have considered the idea of an in vivo study, but, in addition to being very expensive, it requires lengthy authorization procedures and from an ethical point of view it is increasingly discouraged. Thus, while we are trying to stabilize mEVs suspension for oral administration in vivo experiments, we will test complex in vitro models.

Reviewer 2 Report

I appreciate the opportunity to review the manuscript for publication in MDPI IJMS. I feel that the topics are interesting and the manuscript is grossly well organized. I find the authors' work to be scientifically valuable, well written and of interest.

I have a few comments as follows.

This study aimed to investigate the immunomodulatory effects of goat milk extracellular vesicles (mEVs) on toll-like receptors (TLRs) and immune genes in a pro-inflammatory environment using a porcine intestinal epithelial cell line (IPEC-J2).

In a series of the Figs, statistical differences are evaluated Control/LPS+mEVs vs LPS). It is more easy to understand if they are evaluated in the style as LPS/LPS+mEVs vs Control.

The authors investigated the anti-inflammatory effect of goat mEVs by means of applying two models that mimic CD (using LPS) [31] and UC (using H2O2) [15]. It would be desirable to present the difference in downstream inflammatory processes between the two stimuli.

Additional piece of graphical abstract would be nice to present the summary, since the manuscript contains loads of interesting contents.

Author Response

Thanks to the referee for the suggestion

This study aimed to investigate the immunomodulatory effects of goat milk extracellular vesicles (mEVs) on toll-like receptors (TLRs) and immune genes in a pro-inflammatory environment using a porcine intestinal epithelial cell line (IPEC-J2).

In a series of the Figs, statistical differences are evaluated Control/LPS+mEVs vs LPS). It is more easy to understand if they are evaluated in the style as LPS/LPS+mEVs vs Control.

We thank the reviewer for taking the time to evaluate our paper. We have decided to show the results focusing attention on LPS because we wanted to demonstrate the effect of mEVs+LPS vs LPS and the effect LPS vs Control. In the first case we demonstrated the mEVs effect on inflammation and in the second case the inflammatory effects of LPS.

The authors investigated the anti-inflammatory effect of goat mEVs by means of applying two models that mimic CD (using LPS) [31] and UC (using H2O2) [15]. It would be desirable to present the difference in downstream inflammatory processes between the two stimuli.

Thanks to the referee for the suggestion. We have added some information, see lines 118-124

Additional piece of graphical abstract would be nice to present the summary, since the manuscript contains loads of interesting contents.

Reply. A graphical abstract was added.
